# Co-Infections, Secondary Infections, and Antimicrobial Use in Patients Hospitalized with COVID-19 during the First Five Waves of the Pandemic in Pakistan; Findings and Implications

**DOI:** 10.3390/antibiotics11060789

**Published:** 2022-06-09

**Authors:** Kiran Ramzan, Sameen Shafiq, Iqra Raees, Zia Ul Mustafa, Muhammad Salman, Amer Hayat Khan, Johanna C. Meyer, Brian Godman

**Affiliations:** 1Department of Medicine, Allama Iqbal Medical College, Lahore 54000, Pakistan; kiranramzan1994@gmail.com; 2Department of Medicine, Faisalabad Medical University, Faisalabad 38000, Pakistan; s4sameen09@yahoo.com (S.S.); iqraraees2013@gmail.com (I.R.); 3Discipline of Clinical Pharmacy, School of Pharmaceutical Sciences, Universiti Sains Malaysia, Gelugor 11800, Penang, Malaysia; dramer2006@gmail.com; 4Department of Pharmacy Services, District Headquarter (DHQ) Hospital, Pakpattan 57400, Pakistan; 5Department of Pharmacy, University of Lahore, Lahore 54000, Pakistan; msk5012@gmail.com; 6Department of Public Health Pharmacy and Management, School of Pharmacy, Sefako Makgatho Health Sciences University, Pretoria 0204, South Africa; hannelie.meyer@smu.ac.za; 7Centre of Medical and Bio-allied Health Sciences Research, Ajman University, Ajman 346, United Arab Emirates; 8Strathclyde Institute of Pharmacy and Biomedical Science (SIPBS), University of Strathclyde, Glasgow G4 0RE, UK

**Keywords:** COVID-19, successive waves, antimicrobial utilization, antimicrobial resistance, resistance patterns, hospitalized patients, guidelines, Pakistan

## Abstract

Background: COVID-19 patients are typically prescribed antibiotics empirically despite concerns. There is a need to evaluate antibiotic use among hospitalized COVID-19 patients during successive pandemic waves in Pakistan alongside co-infection rates. Methods: A retrospective review of patient records among five tertiary care hospitals during successive waves was conducted. Data were collected from confirmed COVID-19 patients during the first five waves. Results: 3221 patients were included. The majority were male (51.53%), residents from urban areas (56.35%) and aged >50 years (52.06%). Cough, fever and a sore throat were the clinical symptoms in 20.39%, 12.97% and 9.50% of patients, respectively. A total of 23.62% of COVID-19 patients presented with typically mild disease and 45.48% presented with moderate disease. A high prevalence of antibiotic prescribing (89.69%), averaging 1.66 antibiotics per patient despite there only being 1.14% bacterial co-infections and 3.14% secondary infections, was found. Antibiotic use significantly increased with increasing severity, elevated WBCs and CRP levels, a need for oxygen and admittance to the ICU; however, this decreased significantly after the second wave (*p* < 0.001). Commonly prescribed antibiotics were piperacillin plus an enzyme inhibitor (20.66%), azithromycin (17.37%) and meropenem (15.45%). Common pathogens were *Staphylococcus aureus* (24.19%) and *Streptococcus pneumoniae* (20.96%). The majority of the prescribed antibiotics (93.35%) were from the WHO’s “Watch” category. Conclusions: Excessive prescribing of antibiotics is still occurring among COVID-19 patients in Pakistan; however, rates are reducing. Urgent measures are needed for further reductions.

## 1. Introduction

In Pakistan, the first laboratory-confirmed case of a novel coronavirus, subsequently referred to as COVID-19, was reported on 26 February 2020. Following this, an appreciable number of positive cases were reported throughout the country in successive waves [1,2], which is similar to other countries [3,4]. The first wave of COVID-19 in Pakistan peaked on 14 June 2020, infecting more than 300,000 people [5,6]. However, due to strict lockdown activities, travel restrictions, enhanced capacity testing for COVID-19 as well as improved contact tracing and isolation, cases started to decline substantially, mirroring other countries [5,7,8,9]. The second wave of COVID-19 was declared by the Government of Pakistan on 28 October 2020, with greater infection, positivity and mortality rates [10,11]. The third COVID-19 wave impacted mainly the Punjab and Khyber Pakhtunkhwa provinces of Pakistan, with the peak of cases in this wave occurring in April 2021 [12,13]. The fourth wave of COVID-19 was declared in Pakistan in July 2021, with the highest number of positive cases reported in August 2021, and a positivity rate of 6.78% [14]. According to the official statement from the National Command and Operation Center (NCOC), Government of Pakistan, the omicron-driven fifth COVID-19 wave peaked on 23 January 2022 with 7586 positive cases reported in 24 h [15]. More than 1,520,817 confirmed cases and 30,326 deaths have been reported in Pakistan as of 21 March 2022 [16]. On 21 March 2022, in the last 24 h, 1,476,120 patients had recovered, with 361 positive cases reported [17].

While most patients with COVID-19 in Pakistan manifest with mild to moderate disease, severe cases still typically required hospitalization as well as mechanical ventilation [18,19]. Since the start of the pandemic, numerous pharmacological options have been proposed to treat hospitalized COVID-19 patients. These include antipyretics, antihistamines, anticoagulants, corticosteroids, immunomodulatory agents, hydroxychloroquine, and antivirals, including remdesivir, and ivermectin, with the most evidence for corticosteroids [20,21,22,23]. There were concerns about the safety of hydroxychloroquine with deaths reported [24,25]. Subsequent studies, including systematic reviews, failed to find clinical benefits from the use of hydroxychloroquine, lopinavir/ritonavir, remdesivir and ivermectin, which led to their removal from the World Health Organization’s (WHO) and other guidelines [26,27,28,29,30,31,32]. Antibiotics are usually not recommended prophylactically in patients with COVID-19 as this is a viral disease, and should ideally only be prescribed following confirmation of any bacterial co-infection or secondary infections [20,32]. However, a number of studies have documented appreciable prescribing of antibiotics among hospitalized COVID-19 patients, including children, despite only 1.2–14.3% of hospitalized patients being identified as having bacterial co-infections or secondary infections [33,34,35,36,37,38]. 

The excessive and unnecessary use of antibiotics in patients with COVID-19 is a crucial driver of antimicrobial resistance (AMR), compromising global health and food security [39,40]. As a result, there have been concerns about the worsening of AMR during the current pandemic, especially in hospital settings, which urgently need to be addressed [41,42,43,44]. The lack of infection prevention and control measures along with antimicrobial stewardship programs (ASPs), combined with the disruption of surveillance activities in hospitals during the current pandemic, could have a disastrous impact on increasing AMR, especially among low-middle income countries (LMICs), if not addressed [45,46,47]. The antimicrobial resistance collaborators (2022) calculated that 1.27 million deaths had occurred in 2019 due to bacterial AMR, with 4.95 million deaths associated with bacterial AMR globally in 2019 [48], making AMR the next pandemic if not addressed. This is because the impact of AMR will worsen unless addressed with more than 10 million people likely to die annually from AMR by 2050, along with economic losses estimated at $US100 trillion per year, which is equivalent to a 3.8% reduction in annual gross domestic product per country [49,50].

The morbidity, mortality and costs associated with AMR resulted in multiple global, regional and national activities in an attempt to improve future antibiotic use, including those from the WHO [39]. The WHO, in 2015, instigated the Global Action Plan (GAP) against AMR, highlighting the need for countries to develop their own national action plans (NAP) to tackle rising rates [51]. More recently, the WHO developed the AWaRe (Access, Watch and Reserve) classification of antibiotics as a monitoring tool for antimicrobial stewardship (AMS) activities across sectors and countries. Antibiotics in the ‘Access’ group should be used against commonly encountered infections with a lower resistance rate. Antibiotics in the ‘Watch’ group should only be used in critical conditions, as they have a greater chance of resistance development, while those in the ‘Reserve’ group should only be prescribed in multi-drug resistance cases [52,53]. The aim is to curb rising AMR rates, especially surrounding ‘Watch’ and ‘Reserve’ antibiotics [52,53,54,55], with ongoing activities across countries to monitor antibiotic prescribing using these criteria as part of the national or local quality improvement programmes [56,57,58].

Pakistan developed its national action plan against AMR in 2017. Strategic priorities included the establishment of robust antimicrobial surveillance, regulations of appropriate antimicrobial utilization and estimations of the costs of AMR [59,60]. However, there are ongoing challenges with the implementation of the NAP in Pakistan that need addressing going forward [61]. 

Currently, though, surveillance data on antimicrobial use among hospitalized patients, particularly among those with COVID-19 is scarce in Pakistan, with typically paper-based systems and limited resources in terms of personnel and available finances [61]. There have been a few studies, including point prevalence survey (PPS) studies, that have reported antimicrobial use in the early phases of the pandemic [37,62]. Consequently, there is a need to address this to provide future guidance, given the concerns with rising AMR rates in Pakistan. There is also a need to track antimicrobial consumption during successive waves of COVID-19 to monitor trends in their prescribing among hospitalized patients with COVID-19 [63]. In view of this, the objective of this study was to evaluate antimicrobial use among hospitalized COVID-19 patients during the first five successive waves of the pandemic in Pakistan. These findings can be used to guide future activities to reduce unnecessary antimicrobial prescribing among patients in Pakistan who are hospitalized with COVID-19. This includes potential quality improvement programmes, including the instigation of ASPs.

## 2. Results

### 2.1. Demographic Characteristics of COVID-19 Patients

A total of 3221 patients hospitalized with COVID-19 across the first five COVID-19 waves were included in the study (Table 1). The majority of patients were male (51.53%), residents from an urban area (56.35%) and belonged to the age groups >50 years (52.06%) followed by 30–50 years (32.25%). Nearly two-thirds of the COVID-19 patients (63.89%) did not have any comorbidity conditions. The frequent comorbidities that were identified and documented in patients’ records included hypertension (14.25%), diabetes mellitus (10.92%), and heart disease (4.43%).

### 2.2. Clinical Characteristics of COVID-19 Patients

The common clinical symptoms that were identified included coughing (20.39%), fever (12.97), a sore throat (9.50%), headache (10.49%) and dyspnea (6.92%). Overall, 13.78% of surveyed patients manifested multiple symptoms (Table 2). More than one-third of COVID-19 patients (34.11%) had white blood cell (WBCs) values out of range, with abnormal X-ray findings evident in 30.12% of the surveyed patients. Less than a quarter of COVID-19 patients (24.18%) were on oxygen therapy and two-thirds of patients (66.02%) stayed in a hospital for 7 to 14 days. The majority of patients (81.83%) were admitted to the medical unit of hospitals, followed by the intensive care unit (18.16%). Most of the COVID-19 patients (45.48%) manifested moderate symptoms while 23.62% presented with mild symptoms. The vast majority of patients were discharged (96.98%) with 3.01% dying from the virus (Table 2), with similar rates across successive waves.

### 2.3. Antimicrobials Prescribed to COVID-19 Patients during Different Waves

An appreciable number of COVID-19 patients (89.69%) were prescribed antibiotics during their hospital stay, even though only a few of those prescribed antibiotics had documented bacterial co-infections (1.14%) or bacterial secondary infections (3.14%) (Table 3). Out of the total number of patients surveyed, 2889 patients hospitalized with COVID-19 were prescribed 5565 antibiotics, with an average of 1.66 ± 0.85 antibiotics per patient.

In both the first and second waves, there was a high prevalence of prescribed antibiotics at an average of 2.03 ± 1.01 and 2.07 ± 0.87 antibiotics/patient, compared with the fourth and fifth waves at an average of 1.35 ± 0.64 and 1.23 ± 0.81 antibiotics/patient, respectively. Most of the antibiotics (84.43%) were prescribed at the time of hospital admission. However, among the patients identified with a bacterial co-infection or secondary infections, antibiotics were typically prescribed after confirmation from the culture results. Overall, 36.62% of surveyed patients hospitalized with COVID-19 were prescribed one antibiotic, 34.12% two antibiotics and 29.24% prescribed three or more antibiotics. The majority (64.81%) of COVID-19 patients were prescribed antibiotics for 6–10 days while 26.63% of patients were prescribed antibiotics for between 1–5 days. Other antimicrobials prescribed included antivirals (remdesivir, 10.28%), antifungals (2.44%) and antiprotozoals (0.66%). 

### 2.4. Association between Antibiotic Use and Demographics as Well as Clinical Characteristics

The differences in antibiotic use among patients based on their demographic and clinical characteristics are shown in Table 4. The utilization of antibiotics (average number of antibiotics per patient) was significantly higher among male patients with COVID-19 (*p* = 0.043) compared to females. There was no significant difference in antibiotics utilization between patients hospitalized during the first and second waves of COVID-19. However, antibiotic utilization was significantly reduced in the three later waves (*p* < 0.001). As expected, antibiotic prescribing increased significantly with increasing severity of COVID-19 (*p* < 0.001), elevated WBCs (*p* < 0.001) and CRP levels (*p* < 0.001) as well as the need for oxygen therapy (*p* < 0.001). Furthermore, antibiotic utilization was also significantly higher among patients initially admitted to the ICU (*p* < 0.001).

### 2.5. Bacterial Agents Identified as Co-Infection and Secondary Infection

Amongst patients with bacterial co-infections along with COVID-19, common pathogens included *Staphylococcus aureus* (42.42%), *Streptococcus pneumoniae* (27.27%) and *Haemophilus influenzae* (24.24%) (Table 5). *Pseudomonas aeruginosa* (24.19%), *Streptococcus pneumoniae* (20.96%) and *Staphylococcus aureus* (19.35%) were frequent among those with secondary bacterial infections.

### 2.6. Prescribed Antibiotics during the Different Waves according to the ATC Classification

Third-generation cephalosporins were the most frequently prescribed antibiotics in 24.1% of patients hospitalized with COVID-19 who were prescribed antibiotics. Other frequently prescribed antibiotics or groups included piperacillin plus enzyme inhibitors (20.7%), macrolides (17.37%), carbapenems (15.45%) and fluoroquinolones (14.32%) (Table 6).

### 2.7. Antibiotics Prescribed during the Different Waves according to WHO AWaRe Classification

The majority of the prescribed antibiotics (93.4%) during the successive waves belonged to the ‘Watch’ category. Overall, 2.3% of prescribed antibiotics were from the ‘Access’ category while 4.3% were from the ‘Reserve’ category (Figure 1).

## 3. Discussion

We believe this is one of the first studies in Pakistan, and potentially among other LMICs, to document the characteristics and management of patients hospitalized with COVID-19 among the same set of tertiary hospitals during successive waves. There have been studies documenting the profile of patients during early waves in LMICs, but not necessarily documenting their characteristics and management during successive waves [64,65,66]. The most common comorbidities reported include hypertension, diabetes mellitus, heart diseases and respiratory diseases, similar to previous studies reported in Pakistan, China, Germany, France and the USA [19,67,68,69,70]. 

The common symptoms of admitted patients, including coughing, fever, sore throats and headaches, or a combination of these, are also similar to previous studies from Pakistan as well as those from Bangladesh and Iran [19,71,72,73,74]. Our findings, that approximately one-third of patients hospitalized with COVID-19 had abnormal X-ray findings and WBCs counts, with 17.1% manifesting out of range of CRP levels, is similar to a study from Jordan regarding abnormal X-ray findings [75] and China regarding WBCs [76]. This contrasts with previous studies in Pakistan where there was a higher percentage of X-ray abnormalities among hospitalized patients [77,78], as well as studies from India and the USA [79,80]. In addition, in a published study in the USA, almost all hospitalized patients with COVID-19 had raised CRP levels [81]. We are not sure of the reasons behind these differences within and between these countries. This may just reflect the current situation at the time across countries; however, further research is needed before we can say anything with certainty

Our findings that less than a quarter of the total hospitalized COVID-19 patients had mild disease, whilst moderate and severe disease were reported in more than 45 and 16% of patients, respectively, contrasts with a previous study from three hospitals in Punjab where more than one-third of COVID-19 patients had severe disease [82]. This may again reflect the changing nature of the virus, especially with ongoing vaccination programmes, which have shown to be effective in reducing the extent and severity of COVID-19 [83,84,85]. However, further research is again needed in this area, including the impact of the vaccination programmes on the severity of admitted patients, before we can say anything with certainty.

Similar to a previous study conducted in the UK, more than two-thirds of patients stayed in hospitals for 7–14 days with more than 20% of patients staying for ≥ 15 days [86]. This was lower than a systematic review published in 2020 [87], which may again reflect greater knowledge in successfully treating patients with COVID-19 in recent years as well as reduced severity with the increasing availability of vaccines. Our findings that 18% of hospitalized patients with COVID-19 required an ICU stay is similar to studies from the USA [88,89]. This contrasts though with a systematic review and meta-analysis published in 2020, where approximately one-third of patients with COVID-19 were admitted to the ICU during hospitalization [90]. 

Encouragingly, the vast majority of patients (~97%) in our study were discharged from hospital. This contrasts with previous studies from Pakistan as well as from China, India, Iran, Poland and the USA, with lower rates [82,91,92,93,94,95,96]. These combined findings may again reflect greater knowledge in successfully treating patients with COVID-19 in successive waves.

Of concern, however, was the high prevalence (~90%) of antibiotic prescribing in our study, particularly broad-spectrum antibiotics, despite a very low prevalence of bacterial co-infections and secondary infections. This, though, is in line with previous studies from Pakistan [37,38], a study from China during the first COVID-19 wave [96], and patients with COVID-19 in an ICU in Kosovo [97]. However, studies from the USA and other countries have reported lower figures, with only two-thirds of hospitalized COVID-19 patients prescribed antibiotics [98,99,100]. Similar to studies from Bangladesh and India, more than one-third of patients in our study received one antibiotic, whilst two-thirds of patients were prescribed multiple antibiotics during their hospitalization [101,102]. This high rate of prescribing is a concern for the future, as this will continue to increase AMR rates in the country. Alongside this, piperacillin plus an enzyme inhibitor, azithromycin, meropenem, ceftriaxone and moxifloxacin were the top five frequently prescribed antibiotics in our study, similar to previous studies from Pakistan [37,38]. Other countries have also documented high prescribing rates of broad-spectrum antibiotics among patients hospitalized with COVID-19 [102,103]. In contrast, a study conducted in Scotland reported amoxicillin, doxycycline and co-amoxiclav were the top three antibiotics prescribed among patients hospitalized with COVID-19 [104]. Of equal concern, is that most of the antibiotics prescribed in our study were from the ‘Watch’ group followed by the ‘Reserve’ group’, although this was similar to studies conducted in Africa, Bangladesh and India [102,105,106].

Mirroring the findings from a study in the US, most of the antibiotics prescribed in our study were administered at admission [107]. This contrasts with a study from Scotland where only two-thirds of the antibiotics prescribed were on the first day of hospital admission [104].

The over-prescribing of antibiotics in our study, especially on admission, is emphasized by very few patients actually having bacterial co-infections or secondary bacterial infections at 1.14% and 3.14%, respectively, with *Staphylococcus aureus* and *Streptococcus pneumoniae*, similar to other studies [102,108]. Having said this, the extent of antibiotics prescribed per patient significantly decreased after the second wave. This again may reflect increasing knowledge about managing patients hospitalized with COVID-19; however, there are still concerns with the extent to which COVID-19 patients are being prescribed even one antibiotic in our study.

ASPs and other measures are urgently needed among hospitals in Pakistan to address the unnecessary antimicrobial prescribing, especially among patients admitted with COVID-19 [40,109,110,111,112]. There have been concerns with the instigation of ASPs in LMICs with resources and other issues; however, this is beginning to change [109,110,113,114,115,116]. This is important for Pakistan, given the rising AMR rates and ongoing challenges with the implementation of their NAP [59,60,61].

Consequently, a multi-sectorial engagement of clinicians, pharmacists and other key groups within hospitals, including infection, prevention and control groups, is essential to formulate hospital-based ASPs. This can start with the instigation and monitoring of guidelines among initially tertiary hospitals in the region regarding the appropriate use of antibiotics on admission based on patient diagnosis, local resistance patterns and the WHO’s AWaRe list [40,110,111,117,118,119]. Such activities should reduce inappropriate empiric prescribing of antibiotics on admission. Alongside this, there should be enhanced microbiological testing capacities, starting with tertiary hospitals. This builds on increased testing of patients during the current pandemic, with the findings documented in patients’ notes to guide future antimicrobial prescribing. In addition, measures should be instigated, including education and monitoring, to enhance the early transition from intravenous to oral therapy where appropriate, alongside their discontinuation if no bacterial infections are detected among COVID-19 patients [111]. Patient review dates, as well as start and stop dates, should also be documented in patients’ notes to assess their continued appropriateness as more microbiological data becomes available. These quality improvement initiatives can be part of future ASPs.

Alongside this, there should be repeated surveillance on antibiotic use generally among patients hospitalized with COVID-19 in the surveyed hospitals in Pakistan to document key changes, thus, building on our initial findings. We will also research key factors associated with bacterial co-infections and secondary infections in patients hospitalized with COVID-19 among the surveyed hospitals to further improve appropriate management and discourage inappropriate antimicrobial prescribing.

We are aware of a number of limitations of our study. Firstly, we only included five tertiary care hospitals within the Punjab Province of Pakistan. Secondly, we were unable to record other patient and clinical characteristics apart from those recorded in the study, since there was no proper documentation in the available medical records. However, this is similar to the drawbacks of point prevalence studies, generally. We also did not research the key factors associated with bacterial co-infections and secondary infections, as this was outside the scope of the study. However, this will be a consideration for the future. Similarly, we did not record the AMR patterns among participating hospitals prior to the pandemic, as this was outside the objectives of the study. Despite these concerns, we believe this is one of the first studies conducted in Pakistan to evaluate antibiotics use among patients hospitalized with COVID-19 during the first five waves of the pandemic with robust findings. Consequently, we believe our findings provide insight to improve future antimicrobial prescribing among clinicians, public health experts and policymakers during future waves of the pandemic.

## 4. Materials and Methods

### 4.1. Study Settings and Design

This retrospective medical record review study was conducted among the COVID-19 wards of five tertiary care/teaching hospitals in the Province of Punjab. Punjab was selected for the purpose of this study, as it is the most populous province in Pakistan [58]. Tertiary hospitals were specifically included as they are likely to provide guidance to other hospitals, including secondary and primary hospitals. The medical records of patients admitted over a period of one month during each of the five COVID-19 waves, i.e., June 2020, November 2020, April 2021, August 2021, and January 2022, were reviewed retrospectively.

### 4.2. Study Variables

Based on previous studies [32,52,55,58,70,71,72,78,82,108,120], data on the following variables were recorded from the individual medical records of all patients, hospitalized for COVID-19:Demographic characteristics include the patients’ ages, their sex, residence and presence or absence of any comorbidities, including diabetes mellitus, hypertension and other respiratory diseases. The age distribution categories, i.e., 10–30 years, 31–50 and >50 years, were based on previous studies by the co-authors.Clinical symptoms include a fever, cough, sore throat or headache.Laboratory findings, including X-rays, white blood cell counts (WBC) and C-reactive protein (CRP) were documented. The X-ray findings were reviewed by medical doctors and the treating physician was consulted in case of any confusion. Normal ranges of WBCs and CRP were taken from the reference mentioned on the testing kits.Whether hospitalized COVID-19 patients were on oxygen therapy or not.Ward subspecialty, including medical wards or intensive care units (ICU) on admission.Duration of hospital stay in days.Status of COVID-19 severity, categorized as asymptomatic, mild, moderate, severe or critical. These were categorized as per the guidelines issued by the Ministry of National Health Services, Regulation and Coordination, Government of Pakistan.Outcomes include whether patients were discharged from a hospital or died.Details about the antibiotics prescribed. This includes how many hospitalized COVID-19 patients were prescribed antibiotics during their stay in hospitals, as well as the presence of bacterial co-infection and bacterial secondary infections. Antibiotics were further classified according to the ATC classification as well as the WHO AWaRe classification.Bacterial co-infection was identified as those bacterial infections identified in ≤2 days after hospital admission due to COVID-19, and bacterial secondary infection as bacterial infections identified in >2 days after admission, microbiologically.The total number of antibiotics, the average number of antibiotics per patient, the duration of antibiotic therapy and the consumption of other antimicrobials, including antivirals, antifungal and antiprotozoal antimicrobials.

### 4.3. Data Collection Procedures

A team of investigators retrospectively collected the data from the COVID-19 wards of the five participating hospitals between January and February 2022. The medical records of inpatients during the months of the five waves of the COVID-19 pandemic (June 2020, November 2020, April 2021, August 2021, and January 2022) were reviewed retrospectively to obtain the required information. Data collection was facilitated and overseen by the principal investigator (ZUM) after all investigators were thoroughly trained on the data collection procedures. All data were recorded on a specifically designed data collection form. Clinical staff were only approached where clarity was needed in terms of accessing records or any anomalies in the patient records. This included any confusion regarding X-ray findings.

### 4.4. Inclusion and Exclusion Criteria

All patients who were admitted into the COVID-19 wards of the selected tertiary hospitals during the first five different waves of the pandemic were included in the study. Patients who were not admitted, isolated at home, or admitted at a time other than previously mentioned were excluded from the study.

### 4.5. Statistical Analysis

All data were entered in Microsoft Excel and, after the appropriate coding, were imported into SPSS, version 22, for analysis. Continuous data were expressed as means and standard deviations, whereas categorical data were presented as frequency and percentages. The Student’s t-test was used to evaluate the difference between the two groups, whereas one-way ANOVA was performed for ≥3 groups. To determine the significance between the intergroup variables, Tukey’s honestly significant difference test and Games–Howell post hoc tests were conducted, where applicable. An alpha (*p*) value of less than 0.05 was considered statistically significant.

### 4.6. Ethical Considerations

Ethical approval for the study was obtained from the Human Research Ethics Committee, Department of Pharmacy Practice, The University of Lahore (REC/DPP/FOP/46). Permission to conduct the study in the different hospitals was obtained from the administrators of each hospital prior to data collection.

Patients were not approached to provide informed consent, since this was a retrospective study based on data collected from patients’ medical records with no direct contact with them. This is in line with previous PPS studies undertaken by the co-authors [36,58,121,122,123,124,125].

No personal patient information was collected, and all patient data was kept confidential. All patients had an anonymized study identifier, which was kept confidential and separate from the data, for the purpose of verifying the accuracy of recorded data, where concerns were identified during data cleaning.

## 5. Conclusions and Next Steps

There was excessive prescribing of antibiotics among COVID-19 patients admitted to tertiary hospitals in Pakistan during the first five waves of COVID-19. This was despite a very low prevalence of bacterial co-infections and secondary infections. Whilst the average number of antibiotics prescribed per patient decreased in later waves, there were still high rates of inappropriate prescribing, which needs to be addressed moving forward. Future activities include the instigation of institutional guidelines with the help of key stakeholder groups alongside monitoring their adherence. This can be achieved with the instigation of ASPs, starting in tertiary hospitals where these do not currently exist. Key targets for ASPs also include encouraging greater prescribing of the WHO ‘Access’ group of antibiotics as well as encouraging greater documentation of start and stop dates for antibiotic prescribing in patients’ notes alongside their rationale. Encouraging a de-escalation from IV to oral antibiotics where appropriate is another key target. We will be monitoring these activities to improve future antibiotic prescribing among tertiary hospitals in Pakistan during future waves of this and other pandemics.

## Figures and Tables

**Figure 1 antibiotics-11-00789-f001:**
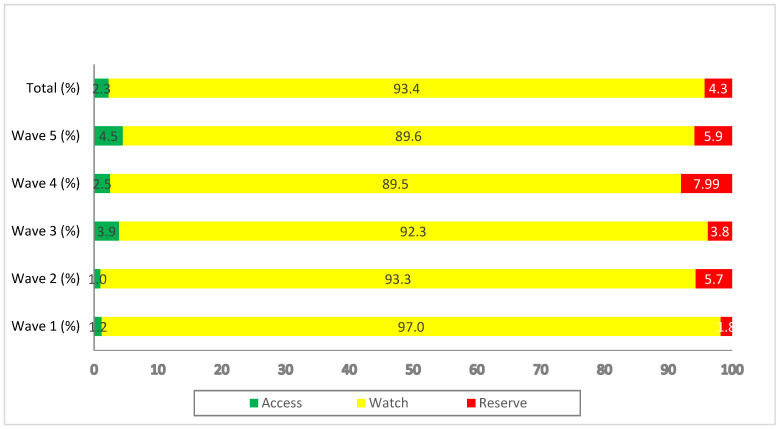
Antibiotics prescribed during the different waves of COVID-19, categorized according to the WHO AWaRe classification.

**Table 1 antibiotics-11-00789-t001:** Demographic characteristics of the hospitalized COVID-19 patients during different waves.

Variables	Number (%) of Patients					
	Wave 1	Wave 2	Wave 3	Wave 4	Wave 5	Total
**Hospitals**						
H1	217 (27.2)	181 (25.8)	163 (19.4)	87 (22.8)	162 (32.5)	810 (25.14)
H2	163 (20.4)	165 (23.5)	190 (22.6)	123 (32.2)	92 (18.5)	733 (22.75)
H3	171 (21.4)	134 (19.1)	221 (26.2)	42 (11.0)	124 (24.9)	692 (21.48)
H4	93 (11.6)	151 (21.5)	156 (18.5)	93 (24.3)	52 (10.4)	545 (16.92)
H5	154 (19.3)	70 (10.0)	112 (13.3)	37 (9.7)	68 (13.7)	441 (13.69)
**Sex**						
Male	467 (58.5)	327 (46.6)	457 (54.3)	182 (47.6)	227 (45.6)	1660 (51.53)
Female	331 (41.5)	374 (53.4)	385 (45.7)	200 (52.4)	271 (54.4)	1561 (48.47)
**Residence**						
Rural	346 (43.4)	274 (39.1)	393 (46.7)	179 (46.9)	214 (43.0)	1406 (43.65)
Urban	452 (56.6)	427 (60.9)	449 (53.3)	203 (53.1)	284 (57.0)	1815 (56.35)
**Age (years)**						
10–30	137 (17.2)	96 (13.7)	123 (14.6)	52 (13.6)	97 (19.5)	505 (15.67)
31–50	304 (38.1)	275 (39.2)	238 (28.3)	108 (28.3)	114 (22.9)	1039 (32.25)
>50	357 (44.7)	330 (47.1)	481 (57.1)	222 (58.1)	287 (57.6)	1677 (52.06)
**Comorbidities**						
None	501 (62.8)	484 (69.0)	422 (50.1)	277 (72.5)	374 (75.1)	2058 (63.89)
Diabetes mellitus	109 (13.7)	56 (8.0)	127 (15.1)	34 (8.9)	26 (5.2)	352 (10.92)
Hypertension	79 (9.9)	87 (12.4)	161 (19.1)	58 (15.2)	74 (14.9)	459 (14.25)
Heart diseases	31 (3.9)	22 (3.1)	68 (8.1)	09 (2.4)	13 (2.6)	143 (4.43)
Respiratory diseases	52 (6.5)	14 (2.0)	33 (3.9)	-	08 (1.6)	107 (3.32)
Others	26 (3.3)	38 (5.4)	31 (3.7)	04 (1.0)	03 (0.6)	102 (3.13)

NB: H = hospital.

**Table 2 antibiotics-11-00789-t002:** Clinical characteristics of hospitalized COVID-19 patients in different waves.

Variables	Number (%) of Patients					
	Wave 1	Wave 2	Wave 3	Wave 4	Wave 5	Total
**Common symptoms**						
Cough	153 (19.2)	156 (22.3)	189 (22.4)	56 (14.7)	87 (17.5)	657 (20.39)
Fever	109 (13.7)	98 (14.0)	110 (13.1)	72 (18.8)	48 (9.6)	418 (12.97)
Sore throat	104 (13.0)	74 (10.6)	90 (10.7)	51 (13.4)	14 (2.8)	306 (9.50)
Headache	73 (9.1)	107 (15.3)	133 (15.8)	28 (7.3)	59 (11.8)	338 (10.49)
Dyspnea	68 (8.5)	56 (8.0)	68 (8.1)	15 (3.9)	27 (5.4)	223 (6.92)
Others	163 (20.4)	87 (12.4)	110 (13.1)	86 (22.5)	159 (31.9)	620 (18.99)
Multiple symptoms	91 (11.4)	57 (8.1)	67 (8.0)	74 (19.4)	104 (20.9)	444 (13.78)
No symptoms	37 (4.6)	66 (9.4)	72 (8.9)	-	-	215 (6.67)
**Laboratory findings**						
Abnormal x-ray	287 (36.0)	177 (25.2)	213 (25.3)	93 (24.3)	57 (11.4)	967 (30.02)
Out of range WBCs	267 (33.5)	321 (45.8)	387 (46.0)	132 (34.6)	126 (25.3)	1099 (34.11)
Out of range CRP	151 (18.9)	136 (19.4)	157 (18.6)	53 (13.9)	31 (6.2)	552 (17.13)
**Oxygen therapy**						
Yes	187 (23.4)	114 (16.3)	126 (15.0)	79 (20.7)	152 (30.5)	779 (24.18)
No	611 (76.6)	587 (83.7)	716 (85.0)	303 (79.3)	346 (69.5)	2442 (75.82)
**COVID severity**						
Asymptomatic	37 (4.6)	66 (9.4)	72 (8.6)	-	-	215 (6.67)
Mild	146 (18.3)	192 (27.4)	234 (27.8)	140 (36.6)	19 (3.8)	761 (23.62)
Moderate	464 (58.1)	259 (36.9)	285 (33.8)	133 (34.8)	316 (63.5)	1465 (45.48)
Severe	105 (13.2)	125 (17.8)	172 (20.2)	82 (21.5)	106 (21.3)	537 (16.67)
Critical	46 (5.8)	59 (8.4)	79 (9.4)	27 (7.1)	57 (11.4)	243 (7.54)
**Ward Subspecialty**						
Medical ward	701 (87.8)	565 (80.6)	683 (81.1)	324 (84.8)	417 (83.7)	2636 (81.83)
Intensive care unit (ICU)	97 (12.2)	136 (19.4)	159 (18.9)	58 (15.2)	81 (16.3)	585 (18.16)
**Duration of hospital stay**						
<7 days	113 (14.1)	139 (19.8)	193 (22.9)	26 (6.8)	98 (19.7)	569 (17.66)
7–14 days	528 (66.2)	479 (68.3)	566 (67.2)	239 (62.6)	278 (55.8)	2090 (64.88)
≥15 days	157 (19.7)	83 (11.8)	83 (9.9)	117 (30.6)	122 (24.5)	562 (17.44)
**Outcomes**						
Discharged	772 (96.7)	682 (97.3)	809 (96.1)	376 (98.4)	485 (97.4)	3124 (97.0)
Deceased	26 (3.3)	19 (2.7)	33 (3.9)	06 (1.6)	13 (2.6)	97 (3.0)

NB: CRP = C-reactive protein; WBCs = white blood cells.

**Table 3 antibiotics-11-00789-t003:** Detail of prescribed antimicrobials during different waves of COVID-19.

Variables	Wave 1	Wave 2	Wave 3	Wave 4	Wave 5	Total N (%)
**Patients prescribed antibiotics**						
Yes	717 (89.8)	656 (93.5)	769 (91.3)	328 (85.8)	419 (84.13)	2889 (89.7)
No	81 (10.1)	45 (6.4)	73 (8.6)	54 (14.1)	79 (15.8)	332 (10.3)
**Presence of bacterial co-infection ***						
Yes	3	7	17	2	4	33 (1.14)
No	21	45	29	9	13	117 (4.04)
Test not availed	693	604	723	317	402	2739 (94.80)
**Presence of bacterial secondary infection ***						
Yes	13	9	31	17	21	91 (3.14)
No	34	18	56	32	16	156 (5.39)
Test not availed	670	629	682	279	382	2642 (91.45)
**Total number of antibiotics for all patients prescribed antibiotics (2889 patients)**	1618	1454	1369	513	611	5565
**Average number of prescribed antibiotics per patient (Mean ± SD)**	2.03 ± 1.01	2.07 ± 0.87	1.63 ± 0.95	1.35 ± 0.64	1.23 ± 0.81	1.66 ± 0.85
**Initiation time of prescribed antibiotics**						
On admission (day 1)	1359 (83.9)	1222 (84.0)	1182 (86.3)	426 (83.0)	510 (83.4)	4699 (84.43)
After 2–5 days	198 (12.2)	212 (14.58)	108 (7.8)	56 (10.9)	58 (9.4)	632 (11.35)
≥6 days	61 (3.7)	20 (1.3)	79 (5.7)	31 (6.0)	43 (7.0)	234 (4.20)
**Number of antibiotics per patient ***						
One antibiotic	127 (18.1)	104 (15.8)	361 (46.9)	202 (61.5)	264 (63.0)	1058 (36.62)
Two antibiotics	279 (39.8)	306 (46.6)	216 (28.0)	67 (20.4)	118 (28.1)	986 (34.12)
Three or more antibiotics	311 (44.3)	246 (37.5)	192 (24.9)	59 (17.9)	37 (8.8)	845 (29.24)
**Duration of prescribed antibiotic therapy**						
1–5 days	511 (31.5)	368 (25.4)	258 (18.8)	179 (34.8)	166 (27.1)	1482 (26.63)
6–10 days	913 (56.4)	1034 (71.5)	941 (68.7)	316 (61.5)	403 (65.9)	3607 (64.81)
>11 days	194 (11.9)	52 (3.5)	170 (12.4)	18 (3.5)	42 (6.8)	476 (14.77)
**Other anti-infective agents**						
Antiviral	157	172	206	82	44	661 (10.28)
Antifungal	67	28	51	11	18	157 (2.44)
Antiprotozoal	19	7	17	-	-	43 (0.66)

NB: * = those prescribed antibiotics (total of 2889 patients), SD = standard deviation.

**Table 4 antibiotics-11-00789-t004:** Differences in antibiotic usage patterns among selected demographic and clinical variables.

Variable	No. of Antibiotics (Mean ± SD)	*p*-Value	Post-Hoc Analysis
**Age (years)**		0.183 *	--
(a) 10–30	1.69 ± 0.94		
(b) 31–50	1.77 ± 0.97		
(c) >50	1.71 ± 0.93		
**Sex**		0.043	--
(a) Male	1.76 ± 0.97		
(b) Female	1.69 ± 0.93		
**COVID-19 wave**		<0.001	
(a) First	2.03 ± 1.01		
(b) Second	2.07 ± 0.87		a > c (*p* < 0.001), a > d (*p* < 0.001), a > e (*p* < 0.001)
(c) Third	1.63 ± 0.95		b > c (*p* < 0.001), b > d (*p* < 0.001), b > e (*p* < 0.001)
(d) Fourth	1.35 ± 0.64		c > d (*p* < 0.001), c > e (*p* < 0.001)
(e) Fifth	1.23 ± 0.81		
**COVID-19 severity**		<0.001	
(a) Asymptomatic	1.43 ± 1.13		
(b) Mild	1.48 ± 0.97		
(c) Moderate	1.73 ± 0.96		c > a (*p* = 0.009), c > b (*p* < 0.001)
(d) Severe	1.98 ± 0.76		d > a (*p* < 0.001), d > b (*p* < 0.001), d > c (*p* < 0.001)
(e) Critical	2.03 ± 0.85		e > a (*p* < 0.001), e > b (*p* < 0.001), e > c (*p* < 0.001)
**Comorbidity**		<0.001	
(a) None	1.71 ± 0.95		
(b) Diabetes mellitus	1.85 ± 0.97		b > c (*p* = 0.001)
(c) Hypertension	1.58 ± 0.90		
(d) Heart diseases	1.78 ± 0.89		
(e) Respiratory diseases	2.06 ± 1.05		e > a (*p* = 0.012), e > c (*p* < 0.001)
(f) Others	2.00 ± 0.92		f > a (*p* = 0.026), f > c (*p* = 0.001)
**X-Ray**		0.049 *	
(a) Abnormal	1.80 ± 0.97		--
(b) Normal	1.71 ± 0.93		
(c) Not performed	1.70 ± 0.96		
**WBCs**		<0.001	
(a) Elevated	1.82 ± 0.94		
(b) Not-elevated	1.66 ± 1.01		a > b (*p* = 0.003), a > c (*p* < 0.001)
(c) Test not performed	1.67 ± 0.93		
**CRP**		<0.001	
(a) Elevated	1.97 ± 0.87		
(b) Not-elevated	1.82 ± 1.05		a > b (*p* = 0.041), a > c (*p* < 0.001)
(c) Test not performed	1.65 ± 0.94		
**Oxygen therapy**		<0.001	--
(a) Yes	2.05 ± 0.78		
(b) No	1.64 ± 0.97		
**Ward Subspecialty**		<0.001	--
(a) Medical ward	1.67 ± 0.96		
(b) Intensive care unit	2.00 ± 0.85		
**Duration of hospital stay**		<0.001	
(a) <7 days	1.21 ± 0.96		
(b) 7–14 days	1.75 ± 0.94		b > a (*p* < 0.001)
(c) ≥15 days	2.18 ± 0.72		c > a (*p* < 0.001), c > b (*p* < 0.001)

* Post hoc analysis revealed no significant difference (*p* ≥ 0.05).

**Table 5 antibiotics-11-00789-t005:** Bacterial agents identified as bacterial co-infection and bacterial secondary infection.

Bacterial Agent	Number (%) of Patients		
	Identified as Co-Infection (n = 33)	Identified as Secondary Infection (n = 91)	Total (n = 124)
*Staphylococcus aureus*	14 (42.42)	16 (17.58)	30 (24.19)
*Streptococcus pneumoniae*	9 (27.27)	17 (18.68)	26 (20.96)
*Pseudomonas aeruginosa*	-	24 (26.37)	24 (19.35)
*Haemophilus influenzae*	8 (24.24)	11 (12.08)	19 (15.32)
*E. coli*	1 (3.03)	13 (14.28)	14 (11.29)
*Klebsiella species*	1 (3.03)	7 (7.69)	8 (6.45)
Other	-	3 (3.29)	3 (2.41)

**Table 6 antibiotics-11-00789-t006:** Prescribed antibiotics during different waves of COVID-19 according to their ATC classification.

ATC Class	Name of Antibiotics (ATC Code)	Wave 1	Wave 2	Wave 3	Wave 4	Wave5	Total (%) (n = 5566)
Third-generation cephalosporin	Ceftriaxone (J01DD04)	255	274	146	33	71	779 (13.99)
	Cephoperazone+ beta-lactamase inhibitor (J01DD12)	183	78	37	86	123	507 (9.11)
Piperacillin and enzyme inhibitor	Piperacillin + enzyme inhibitor (J01CR05)	397	248	311	68	126	1150 (20.66)
Macrolides	Azithromycin (J01FA10)	361	212	271	77	46	967 (17.37)
Carbapenems	Meropenem (J01DH02)	238	279	206	64	73	860 (15.45)
Fluoroquinolones	Ciprofloxacin (J01MA02)	47	77	57	23	24	228 (4.09)
	Levofloxacin (J01MA12)	-	17	12	18	-	47 (0.84)
	Moxifloxacin (J01MA14)	63	156	176	71	57	523 (9.39)
Other antibacterials	Linezolid (J01XX08)	28	83	52	41	36	240 (4.31)
Fourth-generation cephalosporins	Cefepime (J01DE01)	09	15	34	12	22	92 (1.65)
Amoxicillin and beta-lactamase inhibitor	Amoxicillin+ Beta-lactamase inhibitors (J01CR02)	03	-	41	13	28	85 (1.52)
Glycopeptide antibacterials	Vancomycin (J01XA01)	17	-	13	07	05	42 (0.75)
Penicillins with extended spectrum	Amoxicillin (J01CA04)	11	07	09	-	-	27 (0.48)
Other	-	06	08	04		-	18 (0.32)

## Data Availability

The datasets used during the current study are available from the corresponding author on reasonable request.

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
