# Peer review of "Co-Infections, Secondary Infections, and Antimicrobial Use in Patients Hospitalized with COVID-19 during the First Five Waves of the Pandemic in Pakistan; Findings and Implications"

_antibiotics, 2022, doi:10.3390/antibiotics11060789_

Round 1

Reviewer 1 Report

This is an interesting and well-designed study on the use of antibiotics in COVID patients in Pakistan.

Some minor notes

-          In tables 3,4 and 6, could you please provide data on the rates of antibiotic prescription among different hospitals? As there may be a center -dependent variation in antibiotic prescriptions and in the type of antibiotics given

-          It would be helpful to have a subgroup analysis among COVID patients that had co-infections and secondary infections compared to those that did not and try to identify, if any, the risk factors for bacterial infection. In this way, the authors may suggest a subgroup of patients that may benefit from antibiotics/

-          Line 77, please substitute “mechanical breathing” with “mechanical ventilation”

Author Response

Comments and Suggestions for Authors

  1. A) This is an interesting and well-designed study on the use of antibiotics in COVID patients in Pakistan.

Author comments: Thank you for your kind words – appreciated.

  1. B) Some minor notes

-          In tables 3,4 and 6, could you please provide data on the rates of antibiotic prescription among different hospitals? As there may be a center -dependent variation in antibiotic prescriptions and in the type of antibiotics given

Author comments: Thank you for this suggestion. However, as mentioned in the methodology (and now emphasized), we undertook this as a Pan-Pakistan PPS study to investigate the combined situation among all participating hospitals during successive ways. This is similar in design to other Pan-country PPS studies that we have been involved with as well as the Global PPS write-ups. We can look at individual hospitals in separate analyses – however, this will be the subject of separate publications. We hope this is acceptable.

-          It would be helpful to have a subgroup analysis among COVID patients that had co-infections and secondary infections compared to those that did not and try to identify, if any, the risk factors for bacterial infection. In this way, the authors may suggest a subgroup of patients that may benefit from antibiotics

Author comments: Thank you for this helpful comment. However – this would be a different focus than the stated objectives of this study and would necessitate separate in-depth analysis. As such – this would be a complete separate and comprehensive study. We have mentioned this as a limitation and hope this is also acceptable to you.

-          Line 77, please substitute “mechanical breathing” with “mechanical ventilation”

Author comments: Thank you – now done.

Reviewer 2 Report

Thank you for providing an opportunity to review this article. The article provides an useful, overall information about COVID-19 management during 5 waves, in a province of the country.

However, few points need explanation; Please see my comments below.

·       What was the number of positive patients in the second to 5th waves? Do you know how many patients were treated at home?

·         In table 1- COPD patients are not presented, however, indicated in the footnote of the table.

·         In Table 3- the presentation of options- yes / No; are not presented in the same sequence. I suggest presenting it in the same sequence.

·         It is not clear if antibiotics started before or after sending samples for culture.

·         How many patients had culture-positive reports?

·         Authors wrote, watch Antibiotics were prescribed more: It might be so that prescribing watch Antibiotics was based on AMR therefore, it is important to know, what was the AMR pattern of the study area? Please add this information.

·         What was the pre-COVID pattern of prescribing Antibiotics in the setting? Was that same of different from the findings of present study?

·         Page 11- line 251: the word ‘may’, is used 2 times in the same sentence. Need modification

·         Page 11- line 252-53: it is not clear what kind of research is required? Please elaborate.

·         Use of words: ‘Thankfully’, ‘fortunately’ for this kind of research study is not suitable. Please replace.

·         Line 278: what kind of concern is mentioned here? Please elaborate.

·         Line 281: same results were observed in previous study conducted at same study settings so what does the present study add-up to the knowledge?

Author Response

Thank you for providing an opportunity to review this article. The article provides a useful, overall information about COVID-19 management during 5 waves, in a province of the country. However, few points need explanation; Please see my comments below.

Author Comment. Thank you for your kind words – this is appreciated. We hope we have adequately addressed the good points you have raised.

  1. A) What was the number of positive patients in the second to 5thwaves? Do you know how many patients were treated at home?

Author comment. Thank you for the comment. In wave 2, around 750 patients were reported to be positive with COVID-19 in a single day (new reference 11). In Pakistan, 978,662 individuals had been infected with SARS-CoV-2 and 22,642 of them had died up until 15 June 2021 (Wave 3) (new reference 12). As of 31 August, there were 1,160,119 confirmed COVID-19 cases and 25,788 deaths reported, with a positivity rate of 6.78% in the country (new reference 13). We are unsure about the number of patients treated at home – and unsure if there are any reliable data on this. In addition – since we were only interested in the management of patients when hospitalised – especially the use of antimicrobials given increasing concerns with AMR in the country – we did not pursue this further. We hope this is acceptable to you.  

  1. B) In table 1- COPD patients are not presented, however, indicated in the footnote of the table.

Author Comment. Thank you for the correction. It was written by mistake and we have corrected it now. We hope this is now OK.

  1. C) In Table 3- the presentation of options- yes / No; are not presented in the same sequence. I suggest presenting it in the same sequence.

Author comment. Thank you for this comment. As per your comment, we have updated table 3. We hope it is now acceptable.

  1. D) It is not clear if antibiotics started before or after sending samples for culture.

Author comment: Thank you for the comment. We have now upgraded the manuscript under 2.3 to state that most of the antibiotics (84.43%) were prescribed at the time of hospital admission – however, among patients identified with bacterial co-infection or secondary infections these were typically prescribed after confirmation from culture results. We hope this is now acceptable.

  1. E) How many patients had culture-positive reports?

Author comment: Thank you for your comment. The detail about the culture positive tests have been provided in Table 3 along with patients identified with bacterial co-infections or bacterial secondary infection. We hope this is now OK.

  1. F) Authors wrote, watch Antibiotics were prescribed more: It might be so that prescribing watch Antibiotics was based on AMR therefore, it is important to know, what was the AMR pattern of the study area? Please add this information

Author comment: Thank you for this. However – since only very few patients had bacterial co-infections or bacterial secondary infection – very few patients should have been prescribed antibiotics in the first place. In addition – as stated above – those patients with bacterial co-infections or bacterial secondary infection only had antibiotics prescribed following culture reports. Consequently – in view of the main objectives of our study – we did not describe resistance patterns in the participating hospitals before the start of the pandemic – and this was taken into account among the small minority of patients presenting with either bacterial co-infections or bacterial secondary infection. We hope this is acceptable. We have though added this in as a limitation.

  1. G) What was the pre-COVID pattern of prescribing Antibiotics in the setting? Was that same of different from the findings of present study?

Author comment: Thank you again for this comment. However – as stated above – our concerns were the over use of antimicrobials in patients with essentially viral infections – and did this change with successive waves. In view of this – the prescribing patterns before the pandemic in a wide variety of infections seen in the participating hospitals was not collected as this was not in line with the objectives of the study. We hope this is OK with you.

  1. H) Page 11- line 251: the word ‘may’, is used 2 times in the same sentence. Need modification

Author comment: Thank you – now removed

  1. I) Page 11- line 252-53: it is not clear what kind of research is required? Please elaborate.

Author Comment: Thank you – now addressed.

  1. J) Use of words: ‘Thankfully’, ‘fortunately’ for this kind of research study is not suitable. Please replace.

Author Comment: Thank you- now addressed.

  1. K) Line 278: what kind of concern is mentioned here? Please elaborate.

Author Comment: Thank you – now addressed

  1. L) Line 281: same results were observed in previous study conducted at same study settings so what does the present study add-up to the knowledge?

Author Comment: Thank you – this study emphasises high inappropriate prescribing of broad-spectrum antibiotics in successive waves – although reducing. This again emphasizes the need to active ASPs to reduce future AMR – which we have now added in. We hope this is now acceptable.

Reviewer 3 Report

Abstract (please also amend in the main text):

„COVID-19 patients are typically prescribed antibiotics empirically despite concerns.” this sentence needs to be rephrased and complemented

„record review” please rephrase

why over 56 years was selected as a reference age group?

mild or moderate disease presentation

L37-38: the word significantly is used, but no p-values are presented

L41: pathogens instead of agents

L42: from the WHO „Watch” category

L43-44: this sentence needs to be rephrased and complemented

Introduction

„laboratory-confirmed”

L52: wasnt the first official name „novel coronavirus disease”?

L60: greater positivity and mortality rates: please rephrase

L72: due to the emergence

L75: please also write about the controversies about some of these agents

L95: if you deem it relevant to your study, please add the following reference:

https://www.sciencedirect.com/science/article/pii/S1876034121003403?via%3Dihub

L98: and others, such as…

L114: please describe some hindering factors for implementation

Results

Table 1. the differentiation of the age groups of patients should be based on the WHO standard population age groups; „Comorbidities” instead of „Comorbidity”

Table 2. IN the column of duration of hospital stay, the rows are incorrect

L164-165: think through this sentence whether it is appropriate/correct, or rephrase

subsequently, please do the same for all sentences where antibiotics/patient data were presented

Wouldnt it be more appropriate to present (some of) the findings of Table 4. in the form of figures?

Table 5. how was co-infection and secondary infection identified? please provide the case definitions!

please use 1 instead of 01 etc.

Figure 1. what is the unit of measure? please amend the figure

X-ray findings

L251-253: please provide the implications of the successful/unsuccessful COVID vaccination programmes in regards with AMR.

L298: correct bacterial names

please provide some context to self-medication with antibiotics associated with COVID symptoms

Conclusions:

please provide some kind of clinical implications and a take-home message!

Other: is there a significant differences between the contents of this study and the references no. 37 and no. 38?

Author Response

Comments and Suggestions for Authors 

A) Abstract (please also amend in the main text):

i) „COVID-19 patients are typically prescribed antibiotics empirically despite concerns.” this sentence needs to be rephrased and complemented and record review” please rephrase

Author comment: We have revised the abstract where we can bearing in mind the word restrictions instigated by the Journal. We hope this is now OK.

ii) why over 56 years was selected as a reference age group?

Author comment: Thank you for this. We are a little surprised at this comment because this differs from the abstract. We hope our abstract is now clear

iii) mild or moderate disease presentation

Author comment: Thank you for this – now updated

iv) L37-38: the word significantly is used, but no p-values are presented

Author comment: Thank you for this – now updated

v) L41: pathogens instead of agents

Author comment: Thank you for this – now updated

vi) L42: from the WHO „Watch” category

Author comment: Thank you for this – now updated

vii) L43-44: this sentence needs to be rephrased and complemented

Author comment: Thank you for this – now updated

B) Introduction

i) „laboratory-confirmed”

Author Comment – just stated confirmed

ii) Old L52: wasnt the first official name „novel coronavirus disease”?

Author Comment: Thank you – amended this and subsequently stated that this became known as COVID-19. Hope this is now OK

iii) Old L60: greater positivity and mortality rates: please rephrase

Author Comment: Thank you – now done

iv) Old L72: due to the emergence

Author Comment: Thank you – rephrased to start

v) Old L75: please also write about the controversies about some of these agents

Author Comment: Thank you – now inserted. We hope this is now OK.

vi) Old L95: if you deem it relevant to your study, please add the following reference:

https://www.sciencedirect.com/science/article/pii/S1876034121003403?via%3Dihub

Author Comment: Thank you – now added in at the end of the paper as believed logical here. We hope this is acceptable.

vii) Old L98: and others, such as…

Author Comment: Thank you – now removed others

viii) Old L114: please describe some hindering factors for implementation

Author Comment: Thank you – now addressed

C) Results

i) Table 1. the differentiation of the age groups of patients should be based on the WHO standard population age groups; „Comorbidities” instead of „Comorbidity”

Author comment: Thank you for the comment. The age grouping used in our study was based on previous studies as well as age distribution of the study population – we have emphasized this in the Methodology section. Comorbidity is replaced by comorbidities. We hope this is now acceptable

ii) Table 2. IN the column of duration of hospital stay, the rows are incorrect

Author comment: Thank you. We have updated the rows duration of stay rows in table 2.

iii) Old L164-165: think through this sentence whether it is appropriate/correct, or rephrase

Author comment: Thank you for this. We have rephrased the sentence as per your comment - subsequently, please do the same for all sentences where antibiotics/patient data were presented – thank you done

iv) Wouldnt it be more appropriate to present (some of) the findings of Table 4. in the form of figures? Table 5. how was co-infection and secondary infection identified? please provide the case definitions! please use 1 instead of 01 etc.

Author comment: Table 4 presented analysis of post-hoc tests that we believe will be impossible to fully convey as a figure. We hope you agree. Similarly Table 5 without loosing some of the content – especially surrounding % figures. The identification of bacterial co-infection and secondary infection is mentioned in the Materials and Methods section for your kind consideration. Moreover, the corrections have been made in table 5 as per your comment. We also changed ‘01’ to ‘1’, etc. in the Tables add hope these changes are now acceptable to you.

v) Figure 1. what is the unit of measure? please amend the figure

Author comment: Thank you. We have updated figure 1 along with unit of measure

vi) X-ray findings

Author comment: Thank you – these are discussed in 2.2

vii) Old L251-253: please provide the implications of the successful/unsuccessful COVID vaccination programmes in regards with AMR.

Author Comment: Thank you – now added in with references. We hope this is now OK.

viii) Old L298: correct bacterial names

Author Comment: Now spell checked – hope OK.

ix) please provide some context to self-medication with antibiotics associated with COVID symptoms

Author Comment: Thank you – we and others have published on the extensive purchasing of antibiotics without a prescription generally in Pakistan (and throughout Africa and Asia) – which has been exacerbated by the pandemic. However – we have chosen not to include this data as our emphasis has been on antimicrobial use among tertiary hospitals in Pakistan for the reasons stated. We hope this is OK with you.

D) Conclusions:

please provide some kind of clinical implications and a take-home message

Author comment:  Now added into the Discussion – we hope this is now OK.

Reviewer 4 Report

The manuscript has large scientific content but is poorly written.

Discussion and conclusion should be revised, poorly written

Manuscript should attract readers by including figures; graphs (keep tables as supplementary in some cases)

Recent references in introduction. Reduce the number of references, self-citation should be avoided.

Results: Distribution of age in Table 1 does not match with statement given in line 130-131.

Please discuss scenario in Pakistani population and other LMIC,s

Point 2.4 (line 179) does not discuss any association of demographic factors with antibiotic usage

Please write Table 5 in line 197

Data is not linked to antibiotic usage; discussion is mere repetition of result.

Author Response

1) The manuscript has large scientific content but is poorly written.

Author Comment: Thank you – we have now been through the manuscript and updated it. One of the co-authors is a native English speaker with more that 450 publications in peer-reviewed Journals to his name. We hope this is now acceptable.

 2) Discussion and conclusion should be revised, poorly written

Author Comment: We have revised the Discussion to focus on similarities/ differences from other published studies in line with the normal practice for discussions in Journals. We have also reviewed the conclusion. We hope these changes are now acceptable.

 3) Manuscript should attract readers by including figures; graphs (keep tables as supplementary in some cases).

Author Comment. Thank you for this comment. We have re-looked at our Tables and Figures. However – we believe they all convey essential information describing the situation, etc., in line with our objectives. In addition, based on our considerable experience and attempts – we believe it is difficult to convey the extent of information in the Tables in Figures. We hope this is OK with you.

 4) Recent references in introduction. Reduce the number of references, self-citation should be avoided.

Author Comment: As seen – we have reduced the number of references, etc., in the Introduction as well as the number of self citations unless these are essential to convey the messages. This includes citing examples of PPS studies we have undertaken as there have been concerns with Reviewers in previous PPS studies we have submitted to Antibiotics that clearly did not understand the rationale/ principles of PPS studies – hence we wanted to convey that we clearly understand these through multiple publications in peer-reviewed journals. We hope this is now acceptable to you.

 5) Results: Distribution of age in Table 1 does not match with statement given in line 130-131.

 Author Comment:  Thank you for the comment. We have updated the age distribution statement in the results section as well as highlighted in the Methodology section why these age groups were chosen. We hope this is now OK.

 6) Please discuss scenario in Pakistani population and other LMIC,s

Author Comment: In both the Introduction and Discussion – we have extensively referred to other LMICs including Pakistan. We hope this is now OK with you.

7) Point 2.4 (line 179) does not discuss any association of demographic factors with antibiotic usage.

Author comment: Thank you for the comment. Demographic factors including sex, etc., were found to be significantly associated with antibiotic usage – and these are discussed in the results. We hope this is now OK.

 8) Please write Table 5 in line 197

Author comment: Thank you for the comment. We have made correction as per your comment.

 9) Data is not linked to antibiotic usage; discussion is mere repetition of result.

Author comment: Thank you. We have now refined the Discussion to discuss more how our findings are similar/ different to others and the potential reasons for this. We hope this is now acceptable.

Round 2

Reviewer 4 Report

Thank you for revising your manuscript, it is looking acceptable now.